



# Cloud microphysical measurements at a mountain observatory: comparison between shadowgraph imaging and phase Doppler interferometry

Moein Mohammadi[1], Jakub L. Nowak[1], Guus Bertens[2], Jan Molacek[2], Wojciech Kumala[1], and Szymon P. Malinowski[1]

[1]Institute of Geophysics, Faculty of Physics, University of Warsaw, Warsaw, 02-093, Poland
[2]Max Planck Institute for Dynamics and Self-Organization, Goettingen, 37077, Germany

**Correspondence:** Moein Mohammadi (moein.mohammadi@fuw.edu.pl)

**Abstract.** The microphysical properties of cloud droplets, such as droplet size distribution and droplet number concentration, were studied. A series of field experiments were performed in the summer of 2019 at the Umweltforschungsstation Schneefernerhaus (UFS), an environmental research station located just below the peak of Mount Zugspitze in the German Alps. A VisiSize D30 manufactured by Oxford Laser Ltd., which is a shadowgraph imaging instrument, was utilized for the first time to measure the size and velocity of cloud droplets during this campaign. Furthermore, a phase Doppler interferometer (PDI) device, manufactured by Artium Tech. Inc., was simultaneously measuring cloud droplets. After applying modifications to the built-in software algorithms, the results from the two instruments show reasonable agreement regarding droplet sizing and velocimetry for droplet diameters larger than 13 µm. Moreover, discrepancies were observed concerning the droplet number concentration results, especially with smaller droplet sizes. Further investigation by applying appropriate filters to the data allowed the attribution of the discrepancies to two phenomena: the different optical performance of the sensors with regard to small droplets and high turbulent velocity fluctuations relative to the mean flow that result in an uncertain estimate of the volume of air passing through the PDI probe volume.

## 1 Introduction

The environmental research station Schneefernerhaus (UFS) is Germany's highest research station, constructed on the southern side of Mount Zugspitze in the German Alps at a height of 2650 m. For over 20 years, many different institutions have been working there on a variety of permanent studies on an inter- and transdisciplinary basis (Beck and Neumann, 2020). There have been several recent atmospheric studies, including in situ measurements and remote sensing of aerosols as well as greenhouse gases conducted at the UFS (e.g. Klanner et al., 2018; Ghasemifard et al., 2019; Wang et al., 2019, 2020). It has also been used as the measurement site in several campaigns on cloud research (e.g. Siebert and Teichmann, 2000; Wirth et al., 2012; Risius et al., 2015).

Siebert and Teichmann (2000) investigated the influence of liquid water content (LWC) on the behavior of sonic anemometers through an intercomparison experiment at the UFS, where the results obtained under different conditions (dry/cloudy) give





no significant indication of such an influence. Moreover, Wirth et al. (2012) chose Mount Zugspitze for their systematic observations of banner cloud events. Later, using meteorological data collected by the German Weather Service (DWD) from 2000 to 2012 and turbulence measurements recorded by multiple ultrasonic sensors in 2010, the UFS was introduced as a wellsuited station for cloud-turbulence research, especially for measurements that would be difficult to obtain from an airborne system (Risius et al., 2015; Siebert et al., 2015).

Siebert et al. (2015) used a PDI device to measure cloud droplet size distribution (DSD) and droplet number concentration (DNC). The data showed wide variation in DNC with an almost uniform droplet size, clearly suggesting the prevalence of inhomogeneous mixing. The analysis of data measured at the UFS, such as time series of DNC, LWC, energy dissipation rate, etc., was performed in a manner similar to data recorded in prior airborne measurements (Siebert et al., 2006) to compare the two approaches. Consequently, the results supported the argument that the UFS is a suitable location for detailed Lagrangian measurements of cloud droplets in turbulence.

To measure cloud droplet microphysical properties, different techniques have been used over decades. One common technique is to continuously count droplets one-by-one as they pass a very small active probe volume. This method is also used by a PDI serving as a spectrometer to obtain the droplet size from the interference fringe pattern produced by the scattered light. The PDI has been used for the first time by Bachalo and Houser (1984) for spray droplet size and velocity characterization. Later, Chuang et al. (2008) tested Artium Flight-PDI (F/PDI), as a cloud droplet sizing instrument for airborne measurements. They used a new model for determining the probe volume, then performed inter-comparison based on LWC, DSD, and DNC measurements among instruments: a PDI, PVM-100A (Gerber et al., 1994), and FSSP-100 (Lu et al., 2007). The results showed good agreement between the PDI and the PVM regarding the LWC despite highlighting some differences. However, with respect to the DSD, an approximately 5 μm difference in sizing was revealed between the PDI and the FSSP, in addition to an approximately 20 to 50% difference in spectral breadth (as measured by $D_{90} - D_{10}$). Considering the sizing bias, the DNC results were found to be correlated, but typically differed by a factor of 1.2–2.0, with consistently higher values for the FSSP. This discrepancy in the DNC was believed to be either due to the FSSP triggering noise, resulting in false droplet detections or due to uncertainties in the PDI counting or probe volume.

In contrast to the abovementioned instruments (the FSSP and the PDI), volumetric methods, such as Shadowgraphy and holography, obtain droplet properties from images captured inside a larger sample volume. There are already several studies on cloud microphysics using holography techniques during airborne campaigns (e.g. Fugal and Shaw, 2009; Beals et al., 2015; Glienke et al., 2017; Larsen et al., 2018; Desai et al., 2019). It has also been used on cable car platforms (Beck et al., 2017) and in ground-based experiments (e.g. Raupach et al., 2006; Henneberger et al., 2013; Schlenczek et al., 2017). Ramelli et al. (2020) investigated inhomogeneities in the microphysical properties of stratus clouds by using holography in a tethered balloon system (TBS). The inhomogeneities were characterized by the variability in the DNC and the DSD, which measured up to 200 cm$^{-3}$ and between 6 and 24 μm, respectively. They noticed that the instrument limit in measuring small cloud particles ($D < 6$ μm) could lead to an underestimation of the DNC, especially close to the cloud base, in fog or in clouds with a small mean droplet diameter.



In regard to the implementation of shadowgraph techniques in cloud microphysics, a cloud particle imager (CPI) is considered a useful instrument for airborne measurements. It provides high-resolution (2.3 µm per pixel) shadowgrams of cloud particles over the size range of 2.3–2300 µm (Woods et al., 2018). CPI images have been used to separate water drops from
nonspherical ice particles in mixed-phased clouds (e.g. Lawson et al., 2001; Stith et al., 2002) or identify ice particle habits in cirrus clouds (e.g. Lawson et al., 2010; Woods et al., 2018). However, concerning DSD and DNC measurements in the aforementioned studies, instruments other than a CPI have been mainly used (e.g. PMS-FSSP and PMS-260X in Lawson et al. (2001), 2D-C in Stith et al. (2002), 2D-S and CAS in Lawson et al. (2010), and FCDP in Woods et al. (2018)). In addition to airborne measurements, some interesting laboratory experiments have also been conducted using shadowgraphy to study
droplet collisions in turbulent flows (e.g. Bordás et al., 2013; Bewley et al., 2013) or icing in wind turbines (Rydblom and Thörnberg, 2016). Sijs et al. (2021) recently compared shadowgraph imaging with phase Doppler particle analysis (PDPA) and laser diffraction techniques in spray droplet size measurements. According to their laboratory experiments, the shadowgraph imaging and PDPA techniques agree very well for droplet sizes in the range from 10 µm up to ∼400 µm.

Nowak et al. (2021) investigated applicability of a shadowgraph imaging system called VisiSize D30 for cloud microphys-
ical measurements. The reliability of detection and accuracy of sizing under conditions resembling atmospheric clouds have been verified in this research. The VisiSize D30 (hereafter called VisiSize) was manufactured by Oxford Lasers Ltd.. It works based on a particle/droplet imaging analysis (PDIA) method. PDIA was originally introduced by Kashdan et al. (2003) for the diagnosis of agricultural and industrial sprays. That study was followed by later assessments of PDIA capability for characterization of small diameter, high-speed, two-phase flows (Kashdan et al., 2004). They also compared PDIA with phase
Doppler anemometry (PDA), where excellent agreement was observed between the two techniques for droplets larger than approximately 25 µm in diameter (Kashdan et al., 2007).

In this study, we show the results from the first use of the VisiSize for in situ measurements of cloud droplets. During a field campaign at the UFS observatory on Mount Zugspitze in July and August 2019, over three weeks of measurements, the DSD, DNC, and LWC of warm clouds were measured using the VisiSize alongside a PDI-FPDR probe from Artium Tech. Inc. The
present paper is structured as follows. Section 2 briefly describes the instruments and their measurement principles as well as the modifications applied to the built-in software algorithms to achieve more accurate results. In Section 3, a short description of the field campaign location and instrument setup in addition to the measurement statistics are provided. The results after analyzing the data are provided in detail in Section 4, which is followed by a final discussion and conclusion regarding the application of the VisiSize for cloud microphysics study.

## 85  2  Instrumentation and methods

### 2.1  VisiSize D30 shadowgraph system

The VisiSize is a shadowgraph instrument capable of characterizing particles in various suspensions. A wide range of particle/droplet information, such as droplet size, velocity, and shape provided by the VisiSize makes the device potentially suitable for atmospheric research purposes, especially cloud microphysical measurements. The instrument specifications and proper-



**Table 1.** Overview of the latest shadowgraphy and holography sizing systems for cloud droplets: the VisiSize D30, Cloud Particle Imager (CPI), and HoloBalloon (HOLIMO 3B)

| Instrument | VisiSize D30 | | CPI | HOLIMO 3B |
|---|---|---|---|---|
| Sensor [pix × pix] | 1952×1112 | | 1024×1024 | 5120×5120 |
| Camera Pixel [µm] | 5.5 | | 2.3 | 4.5 |
| Maximum frame rate [fps] | 30 | | 400 | 80 |
| Wavelength [nm] | 808 | | 707 | 355 |
| **Lens setting** | ×2 | ×4 | - | - |
| Lens Magnification | 2.97 | 6.12 | 1.0 | 1.5 |
| Effective Pixel [µm] | 1.85 | 0.9 | 2.3 | 3.0 |
| Resolution [µm] | 3.7 | 2 | 2.3 | 6.0 |
| Field of view [mm×mm] | 3.62×2.06 | 1.75×1.0 | 2.3×2.3 | 15×15 |
| Depth of field [mm] | 16.6 | 5.2 | 7.07 | 100 |
| Sample volume [cm$^3$] | 0.12 | 0.01 | 0.04 | 22.5 |
| Volume rate [cm$^3$ s$^{-1}$] | 3.71 | 0.28 | 16 | 1800 |
| Reference | Nowak et al. (2021) | | Woods et al. (2018) | Ramelli et al. (2020) |

ties for the two camera lens magnification settings recommended by Nowak et al. (2021) for cloud droplet measurements are summarized in Table 1. Moreover, specifications of the CPI and the HoloBalloon (HOLIMO 3B) are also listed in Table 1.

Nowak et al. (2021) have described the droplet detection and sizing system of the VisiSize in detail. In general, the PDIA method, used in the VisiSize, involves illuminating the region of interest from behind with an infrared pulsed laser while collecting shadow images of droplets passing through the measurement volume using a high-resolution camera. Droplets detected inside the depth of field are then measured based on their shadow images, and the size distribution is built by analyzing a series of images. However, some important points have been revealed after laboratory experiments with the VisiSize by Nowak et al. (2021), such as the necessity of developing a correction method for sample volume (SV) calculation or minimum detection size limits for different lens magnifications.

During the DNC calculations, one must always bear in mind that the sample volume (SV) within which a droplet can be detected depends on the droplet size. The reason is that blurring caused by defocus impacts images of droplets of different sizes differently. The built-in software algorithm presumes a linear relation between the depth of field and droplet diameter. The consequent depth of field ($z_{\text{def}}$) forms the "default sample volume" (def. SV) for a droplet $D_i$ which is shown in Figure 1a. Assuming a linear relation between $z_{\text{def}}$ and $D$ is efficient for relatively large droplets ($D > 260$ µm, precise value depends on the lens magnification). However, in the case of small droplets, such as those typically found in clouds, the focus rejection criterion imposes a substantial constraint on an acceptable depth of field ($z_{95}$). Consequently, the focus rejection affects the





relevant SV and results in an underestimated number concentration. Hence, Nowak et al. (2021) developed a correction method using the SV based on a focus rejection limit. It is called the "corrected sample volume" (cor. SV) and is used in this study. Moreover, the length and width of the camera field of view ($L_x$ and $L_y$) are also shown in Figure 1a. The maximum values for $L_x$, $L_y$, and $z_{\mathrm{def}}$ provided by the manufacturer can be found in Table 1.

In addition to the diameter mode used in Nowak et al. (2021), there is also a velocity mode available in the VisiSize software. After adjusting some additional settings (e.g., maximum search radius, maximum deviation angle, and flow direction), droplet size and velocity can be measured simultaneously in velocity mode by using either of the following two methods. "Image pairs" is a method where pairs of consecutive frames are processed, and after detecting a droplet within both images, the droplet velocity is calculated based on its position change between the frames. Another way to measure velocity is called

"double exposed", in which during every camera exposure, two pulses are sent from the laser. Hence, every detected droplet should cast two shadows in each frame. According to the manufacturer, the latter method is not recommended whenever it is possible to use the former to avoid saturation of the camera. Properly adjusting the abovementioned settings in the velocity mode is crucial to obtain accurate velocity. Otherwise, the device may find a false droplet within the second frame or in the same frame as the second shadow (depending on the selected method).

## 120  2.2  PDI-FPDR Probe

Another device named the PDI-FPDR (PDI Flight Probe, Dual Range) was constructed by Artium Technologies, Inc. (hereafter called "PDI") has also been used in the campaign to measure cloud droplets. The PDI is a laser-based diagnostic instrument for simultaneously measuring the size and velocity of individual spherical droplets in polydisperse flow environments. The instrument utilizes the well-known technique of phase Doppler interferometry. The technique is based on the fact that as a

spherical droplet passes through two intersecting beams the phase shift of the light scattered from them is proportional to the droplet size. A theoretical description of the method and details regarding a PDI probe have already been provided in the literature (e.g. Bachalo, 1980; Bachalo and Houser, 1984; Chuang et al., 2008; Bachalo and Sankar, 2016). Hence, we add just a brief explanation in this section.

    A diode-pumped solid-state (DPSS) laser with a 532 nm wavelength is used as the light source for a PDI. The dual size

range version combines two PDI instruments (channels) for small and large size ranges in a compact package. According to the manufacturer, PDI channel 1 is able to measure the size and velocity of droplets from 0.5 to 100 μm, whereas channel 2 can cover up to 1000 μm. The droplet velocity is measured by the method typically used for laser Doppler velocimetry. That is, as a particle crosses the interference fringes created by the two intersecting coherent laser beams, it refracts or reflects the local light intensity onto the receiver. The light intensity pattern projected onto the receiver and directed to the photomultiplier

tubes produces a typical Doppler burst signal whose frequency is directly related to the velocity of the droplet. The droplet magnifies the interference fringe pattern onto the receiver. The degree of magnification is related to the droplet size, and is measured by comparing the spacing of these fringes on the receiver detectors (spatial wavelength) and the fringe spacing. The spatial wavelength is determined by using the phase shift of the signal between the detectors in the time domain along with the calibrated spacing of the detectors.





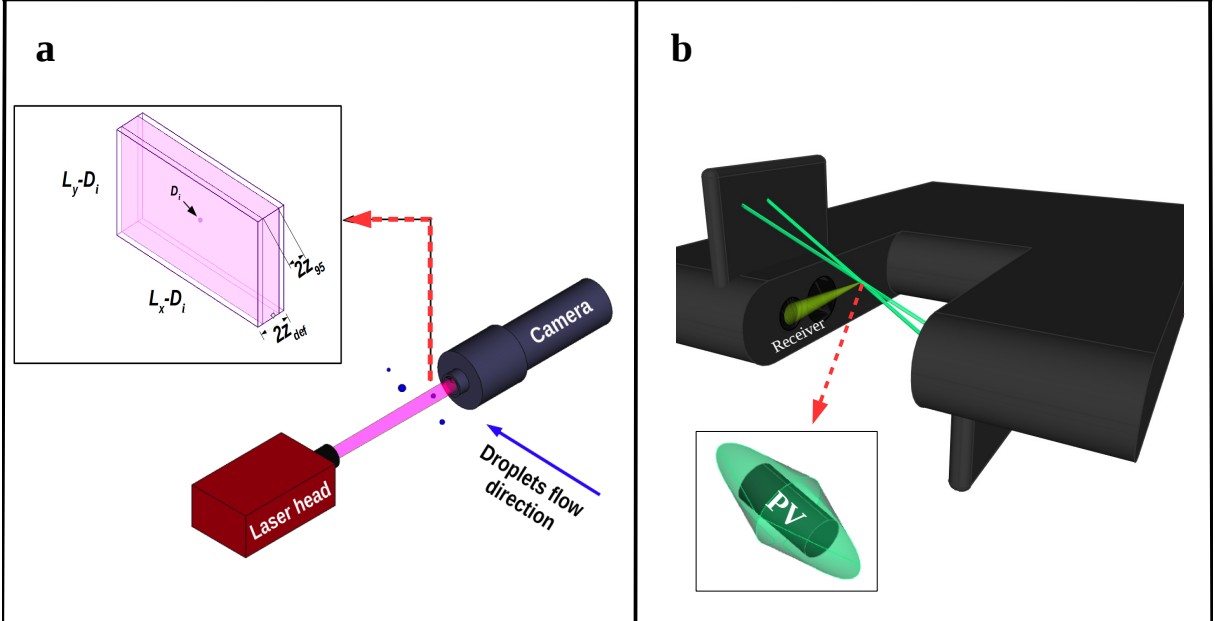

**Figure 1.** Schematic representation of (a): the VisiSize setup and a magnified sample volume (SV) around a droplet, comparing the default and corrected depth of fields ($z_{\text{def}}$ vs. $z_{95}$); (b): a PDI probe with the magnified illustration of the cylindrical probe volume (PV) within the intersection of two laser beams where the light from the passing droplets is scattered into the receiver.

To be measured by a PDI system, a cloud droplet must pass through the probe volume (PV), which is defined by the beam waist of the Gaussian focused laser beam as well as the slit aperture width of the receiver. The shape of the probe volume can be estimated as a cylinder with oblique ends (see Figure 1b). While the cylindrical height of the PV is assumed to be constant, the diameter of the PV base depends on the droplet size. The larger the droplet is, the better it scatters, and therefore the less incident light is required for it to be detected. Hence, taking into account the Gaussian intensity profile of the laser beam, the

PDI has a larger PV for larger droplet sizes. Figure 2 demonstrates how size and position can affect the scattered light from a droplet, and consequently its detection by the PDI. While droplets $D_1$ and $D_2$ have different sizes and positions in the yz plane, both can be detected by the PDI as they pass through their size-dependent PV; in other words, they scatter enough light back to the receiver. However, droplet $D_3$ with the same size as $D_2$, but at a further distance from the yz plane, cannot be detected by the PDI, as it scatters light at lower intensity than minimally required.

Since the PV is size-dependent, the number of droplets in each size class of the distribution is multiplied by a probe volume correction (PVC) factor to reflect the change in sampling volume with droplet size. The PVC factor is simply the ratio of the probe area (PA) for the maximum droplet size bin to the probe area of each individual size bin D. To calculate the probe area, the following formula can be used:

$$PA(D) = \beta \omega \sqrt{log(\frac{D}{D_{\text{min}}})} \, , \tag{1}$$





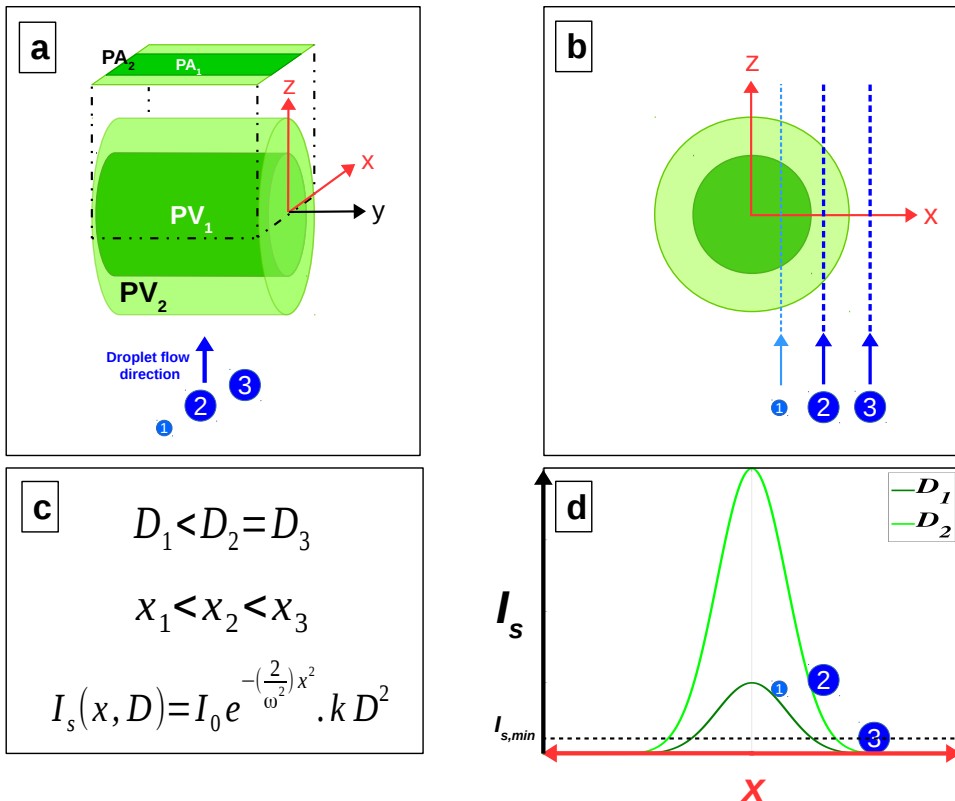

**Figure 2.** (a): Schematic representation of the cylindrical effective probe volume (PV) and probe area (PA) of the PDI for two different sizes of three passing droplets; (b): side view of panel (a) showing trajectories of the droplets located at different positions ($x$) from the $yz$ plane; (c): relationship between size ($D$) and position ($x$) of the droplets along with the formula of the scattered intensity ($I_s$) as a function of $x$ and $D$ (where $k$ is an instrument constant); (d): qualitative graphs of $I_s - x$ illustrating the scattered intensity of each droplet based on its $x$ and $D$ as well as the minimum $I_s$ below which a droplet cannot be detected.

where $\beta$ is a constant based on the aperture and the angle of the detector and is equal to twice the height of the PV, $\omega$ is the laser beam waist, and $D_{\min}$ is the effective minimum diameter for which $PA(D_{\min}) = 0$. Any droplet smaller than $D_{\min}$ will not be detected by the PDI even if it passes through the center of the probe area (with the highest light intensity). The values for $\omega$ and $D_{\min}$ are calculated by the PDI software after each measurement by fitting a model to the raw data.

### 2.2.1  Original DNC methods

There are two methods used by the PDI software to compute the DNC of cloud droplets, which are referred to as the "swept volume" and "transit time" methods in the instrument manual. To be able to distinguish them from the modified methods, which will be introduced later in this section, we call them the original DNC methods ("org. swept" and "org. transit"). In the former



method, the total corrected count (corrected for probe volume) is divided by the swept volume to calculate the DNC and is given as:

$$N_V^{SV} = \frac{1}{(PA)(t_{Tot})} \sum_i \frac{n_{c(i)}}{\overline{|v_i|}} \,, \tag{2}$$

where the average absolute velocity ($\overline{|v_i|}$) is calculated for each size bin $i$ separately to eliminate the velocity biasing on the sampling statistics and is given as:

$$\overline{|v_i|} = \frac{\sum_j |v_{i,j}|}{n} \,, \tag{3}$$

where $v_{i,j}$ refers to the velocity of the $j$-th droplet within the $i$-th size bin ($j$ and $i$ are the droplet and size bin indices, respectively).

With the transit time method, the DNC is determined using the transit times for the droplets as they pass through the sample volume. The transit time is the amount of time that the droplet spends within the probe volume. This method uses the computed ratio of the time duration when there is a droplet in the probe volume, to the total sample time for the measured distribution at that point. The transit time ratio divided by the probe volume yields the DNC and is given by:

$$N_V^{TT} = \frac{1}{t_{Tot}} \sum_i \frac{\sum_j t_{i,j}^T}{PV_i} \,, \tag{4}$$

where $t_{i,j}^T$ refers to the transit time of the $j$-th droplet from the $i$-th size bin. This method is applied for each size class to accommodate the size biasing of the probe volume as a function of droplet size. Both methods rely on an accurate probe area calculation. The transit time method will work best as long as the beams are partially Gaussian at the intersection. The swept volume method works well if the trajectories of the droplets are in the direction of the velocity component measured for a one-component system.

### 2.2.2 Modified DNC methods

As mentioned in Section 2.2.1, calculation of the DNC by the PDI software is based on the average velocity of the droplets passing through the probe volume. Hence, to improve the results reported by the PDI, we modified the swept volume method in two ways to reduce the effect of averaging the velocity as much as possible.

The first approach, which we call the "modified swept volume" (mod. swept), needs to split the long run of measurements into smaller time bins (1 minute or shorter periods). Then, we apply the swept method to each period separately, which allows us to account for changing conditions (e.g., wind velocity). In addition, the bin centers instead of the bin edges are used to calculate the probe area. Another way to modify the swept volume method is to use the interarrival time of each droplet to measure the air volume passing through the probe. Contrary to the default swept volume method, where all droplets within a size bin are assumed to have the same PV, in this approach, which we call the "modified individual" method (mod. individual), each single droplet $j$ has its own swept volume of air, given by:

$$V_j = PA_j |v_j| \tau_j \,, \tag{5}$$





where $PA$ is the probe area for the droplet based on its size, multiplied by droplet velocity and its interarrival time, which is defined below as half of the time period between the detection of the previous and next droplet:

$$\tau_j = \frac{t_{j+1} - t_{j-1}}{2} \ .$$
(6)

Consequently, the DNC is calculated as follows:

$$N_V^{\text{ind}} = \frac{n}{\sum_j V_j} \ .$$
(7)

### 2.2.3 Coincidence filter

Thus far, we have tried to obtain more accurate DNC results for the PDI by considering the changes in the flow velocity and 200 droplet size. This helps to reduce the DNC calculation errors caused by averaging the velocity and size. However, during the measurements, we realized that further corrections are needed. For instance, in theory the PDI should detect droplets passing the probe volume one by one, which means that no more than one droplet can be detected at any specific time. However, within the raw data, we often found droplets with the exact same detection time, which is considered an instrumental error. Hence, we have used a filter to reduce the effect of this error as much as possible. The "coincidence filter" replaces all droplets detected 205 simultaneously with a single droplet, which possesses the average size and velocity of the multiple droplets.

### 2.2.4 $D_{\text{min}}$ cutoff

Another filter that is applied to the raw data is the $D_{\text{min}}$ cutoff. As was mentioned earlier in Eq. 1, the PDI software obtains the $D_{\text{min}}$ value by fitting a model to the raw data after each measurement run. Thus, each time $D_{\text{min}}$ takes a different value below which the PDI cannot reliably measure droplets. The PVC factor for all size bins smaller than $D_{\text{min}}$ is also equal to zero. Hence, 210 their probability distributions in the DSD histogram for the PDI are zero when the PVC factor is applied. However, over the course of our measurements, sometimes the PDI recorded droplets smaller than $D_{\text{min}}$ within the raw data. Therefore, to obtain more accurate results, we set a $D_{\text{min}}$ cutoff on the data, which excludes all droplets smaller than $D_{\text{min}}$ from the analysis. This filter is also applied to the VisiSize data so that it is possible to make a better comparison between the instruments. Note that this is only done for the "modified data" presented in Section 4.

## 3 Mount Zugspitze measurement campaign

Airborne measurements can sometimes be impractical due to the weight or complexity of the setup. However, mountain observatories facilitate in situ measurements for such experiments while providing opportunities for the long-term sampling of cloud properties. The configuration of the VisiSize setup, as shown in detail by Nowak et al. (2021), makes it difficult to perform any airborne measurement. Hence, we also used a mountain observatory for our first field experiment with the VisiSize to measure 220 atmospheric cloud droplets.

The UFS, which was selected for our 3-week campaign in July-August 2019, is located in the German Alps, approximately 300 meters below the peak of Mount Zugspitze (47° 25' 00" N, 10° 58' 46" E), the highest mountain in Germany (2962





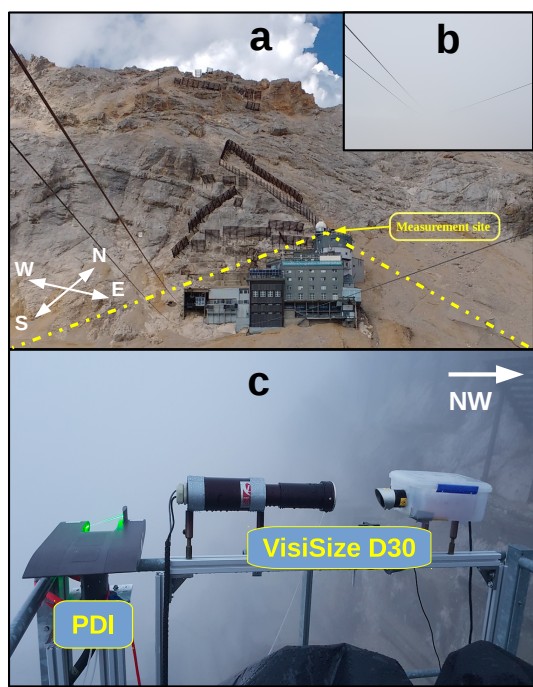

**Figure 3.** Cloud field experiment at the Schneefernerhaus: (a) View of the UFS from Zugspitzplatt on a sunny day, pointing out of the measurement site on the 9th floor platform; (b) The same view while the UFS is immersed in clouds; (c) Main instruments: the VisiSize, and the PDI probe mounted next to each other, while clouds are approaching the UFS from the west side.

m a.s.l.). It consists of 12 floors built into the southern flank of Mount Zugspitze and is frequently immersed in clouds (see Figure 3). As shown in Figure 3c, the main parts of the VisiSize, including the laser head and the camera, were mounted on
an aluminum stand (in a direction 40° to the west of north) alongside the PDI probe. The whole setup was placed on a steel platform, on the western side of the 9th floor rooftop.

During the campaign period, there were nine days with successful cloud event measurements, with both instruments often running simultaneously. We performed several series of measurements with a 15-minute running time. Hereafter, each of these 15 minute long segments is called a "measurement case". Sometimes during the experiment, either one of both instruments did
not function properly. Therefore, to make a comparison later, we excluded all those cases from the analysis. Consequently, 133 cases were selected in which data from both instruments were available over the entire measurement case period. The selected cases include measurements with the VisiSize lens magnifications of ×2 and ×4, as recommended by Nowak et al. (2021), in live diameter or velocity mode. As listed in Table 2, for most of the selected cases, we used the VisiSize ×2 lens magnification in diameter mode. Thanks to the larger sample volume when using this magnification, we could collect as many cloud droplets
as possible during each measurement case.





**Table 2.** Statistics of the VisiSize measurements at the UFS during the 2019 campaign listing the number of selected measurement cases for different modes and lens magnification settings.

| Selected measurements | | 133 | |
|---|---|---|---|
| Lens setting | | $\times 2$ | $\times 4$ |
| VisiSize mode | Diameter | 84 | 26 |
| | Velocity | 18 | 5 |

The PDI probe also measured cloud droplets simultaneously for all selected cases. However, the PDI channel 1 did not work properly during the campaign. It could not collect enough large samples in any of the selected cases (see Appendix for additional information). Hence, for the comparison of the instruments, we only used the data collected with the PDI channel 2 with an adjustment to the sizing range according to the instrument manual. The sizing filter was applied to both instruments so that they could record droplets from the minimum of their detection range (e.g., 1 μm for the PDI channel 2 and 2 μm for the VisiSize mag. $\times 4$) up to 80 μm in diameter. The upper size limit helped to avoid mistakenly recording rain drops in the measurements.

Figure 4 shows the local weather conditions obtained from recordings by the German Weather Service (DWD) at the UFS during the measurements. Each individual point on these diagrams represents a mean of the DWD data measured with a 1 minute time resolution during a selected case (over a 15 minute interval). The average temperature during the measurements typically varied between 0 and 4 °C and the relative humidity usually varied between 95 and 100%. Wind speed was more variable, ranging from ∼1 to over 8 m s$^{-1}$, mostly coming from the southwest direction (between 180 and 270°). The seasonal meteorological conditions as well as the cloud and turbulence properties at the UFS have been published in detail by Siebert et al. (2015) and Risius et al. (2015).

## 4   Results

In this section, we present the results of cloud droplet measurements during the Zugspitze campaign. First, a sample measurement with the VisiSize and the PDI probe is shown. The functionality of both instruments can be compared based on the size range of detected droplets in the sample measurement. Then, we demonstrate the effectiveness of the various additional filters introduced in Section 2 at reducing the discrepancy between the outputs of the two instruments. The presented data are either 1) "raw data" of the detected size and velocity of the droplets; 2) data resulting from the live, internal processing of the instrument software during each measurement (e.g., arithmetic means, DNC, and LWC); or 3) "modified data", which are obtained after applying some or all of the modifications and filters described in Section 2.2 to the raw data. To distinguish between different data processing methods, we use the terms introduced in Section 2: default (def.) or corrected (cor.) sample volume for the VisiSize, and original (org.) or modified (mod.) DNC methods for the PDI.





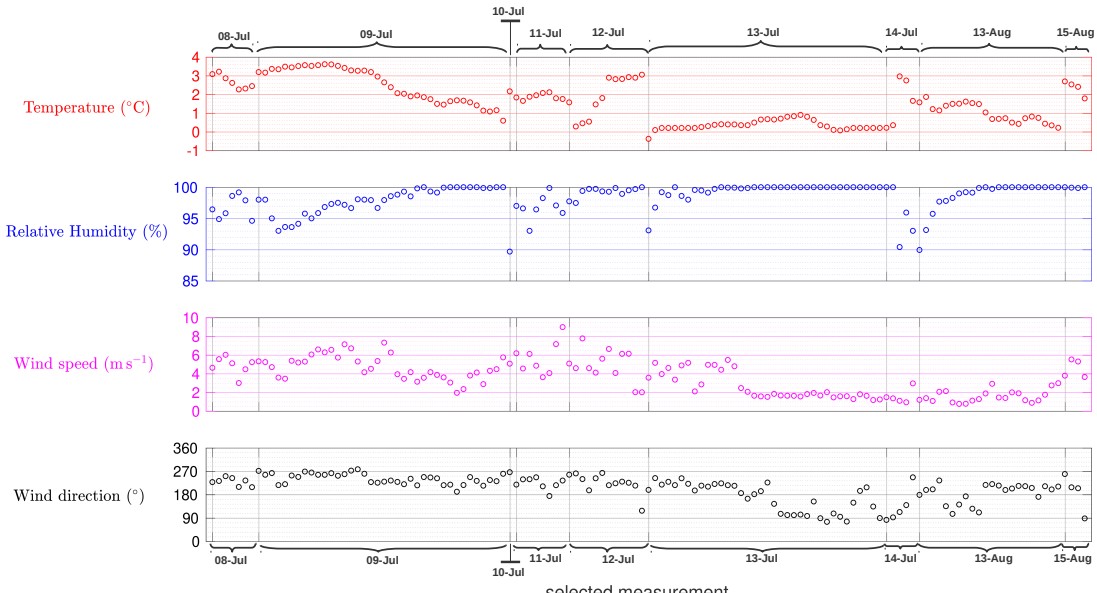

**Figure 4.** Local weather conditions at the UFS, obtained from recordings by the German Weather Service (Deutscher Wetterdienst, DWD) during the measurement campaign including temperature, relative humidity, wind speed and wind direction. Each single point in the plots represents a mean of the DWD data collected during a selected measurement case (over a 15 minute interval). The selected cases on each day are shown in chronological order from left to right. However, two adjacent points do not necessarily represent two consecutive measurements due to the selection process. The wind direction is measured relative to a northerly orientation, with easterly, southerly, and westerly winds assigned $90°$, $180°$, and $270°$ angles, respectively.

## 4.1 Sample measurement case

Table 3 shows the instrument internal raw data processing outputs regarding a measurement conducted on the 13th of July 2019 between 15:36 and 15:51. According to the DWD data (shown in Figure 4), during this cloud event, the wind was blowing with an average speed of $\sim$4.5 m s$^{-1}$ from the southwest (average direction of $\sim$225°), while the average temperature and relative humidity were $\sim$0.4°C and 100%, respectively. A camera lens magnification of $\times 2$ was used for the VisiSize during

this measurement. Different mean diameter values compared in Table 3 are the arithmetic mean ($\bar{D}$), surface mean ($D_2$) and volume mean ($D_3$), which are calculated as follows:





$$\bar{D} = \sum_k \frac{D_k}{n} \,, \tag{8}$$

$$D_2 = \left[ \sum_k \frac{D_k{}^2}{n} \right]^{1/2} \,, \tag{9}$$

$$D_3 = \left[ \sum_k \frac{D_k{}^3}{n} \right]^{1/3} \,, \tag{10}$$

where $D_k$ refers to the diameter of the *k-th* measured droplet and *n* is the total droplet count. Comparing the mean diameter values, we observe an $\sim$3 μm difference between the results of the two instruments in this measurement. The PDI calculates the DNC and the LWC from two different methods (explained in Section 2.2.1), which are listed in Table 3. The DNC value reported by the VisiSize corresponds very well with that obtained from the PDI using the swept volume method. On the other hand, the swept volume method leads to a LWC value from the PDI being twice as high as that from the VisiSize, while they

do agree when using the transit time method.

**Table 3.** Comparison between the cloud droplet measurement data from the VisiSize and PDI probe collected on 13th July 2019 15:36-15:51.

| Instrument | VisiSize ($\times$2 lens mag.) | PDI | |
|---|---|---|---|
| $\bar{D}$ [μm] | 10.23 | 13.08 | |
| $D_2$ [μm] | 10.45 | 13.26 | |
| $D_3$ [μm] | 10.67 | 13.46 | |
| **Method** | - | **transit time** | **swept volume** |
| DNC [cm$^{-3}$] | 348 | 185 | 350 |
| LWC [g m$^{-3}$] | 0.22 | 0.23 | 0.44 |

Figure 5a shows the normalized DSD histogram of the aforementioned measurement for both instruments together with the arithmetic mean diameters displayed in the corresponding legends. In addition to the apparent small μm shift in the distribution, the lack of droplets with diameters below $D_\text{min}$ due to the PVC factor is also clearly visible (see Sections 2.2 and 2.2.4). In Figure (5b), droplets smaller than 9 μm, which is the PDI fit $D_\text{min}$ for this measurement, were filtered out for both instruments,

which resulted in a reduction in the $\bar{D}$ difference. By increasing the minimum cutoff values in the subsequent plots, the DSD histograms, as well as $\bar{D}$ ultimately reach good agreement for droplet sizes larger than 13 μm, as shown in Figure 5d. Hence, we can conclude that the PDI is not as sensitive as the VisiSize in regard to droplets with diameters below 13 μm.





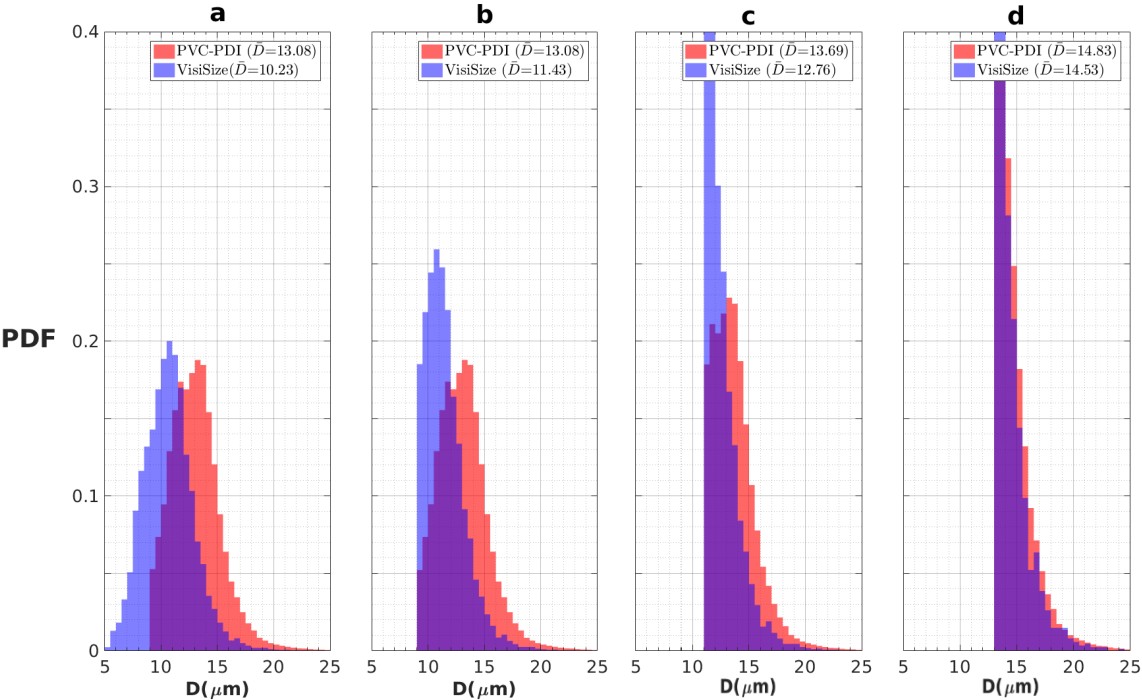

**Figure 5.** Probability distributions of cloud droplet size collected with the VisiSize and the PDI probe during a measurement on 13th July 2019 15:36 – 15:51. The distributions were truncated at progressively large values of droplet diameters for each plot: (a) D>0 µm, (b) D>9 µm, (c) D>11 µm, and (d) D>13 µm. The resulting arithmetic mean diameters for each instrument are shown in the legend.

## 4.2 Sizing

Next, the two instruments are compared based on the mean diameter of cloud droplets measured over the whole campaign
period including all selected measurements. The arithmetic, surface and volume means of droplets measured with the VisiSize and the PDI are shown in Figure 6 from left to right respectively. Each blue circle represents a comparison between the mean diameters obtained from the VisiSize and the PDI based on the raw data of a selected measurement, while the corresponding red circle for that measurement is obtained after applying a modification to the raw data. For the VisiSize the sample volume (SV) was corrected, while the coincidence filter was applied to the PDI raw data as a modification.

The Pearson correlation coefficient (PCC) for each scatter plot is also shown in the legend. Here, the PCC is a normalized version of the covariance showing the strength of the linear relation between the two instruments. It ranges from a perfect negative linear correlation (equal to -1) to a perfect positive linear correlation (equal to 1). Scatter plots for different types of mean diameters in Figure 6 look very similar to each other, with a PCC very close to 1 showing a strong correlation. Mean diameters from the PDI are consistently between 3 and 6 µm higher than those from the VisiSize. Applying modifications to





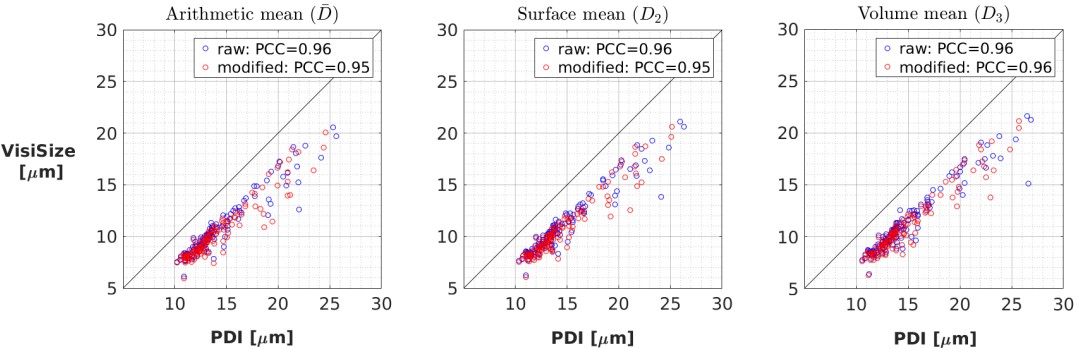

**Figure 6.** Comparison of cloud mean droplet diameter between the VisiSize and the PDI probe. Scatter plots of the arithmetic mean ($\bar{D}$), surface mean ($D_2$) and volume mean ($D_3$) are shown from left to right. Each pair of blue/red circles indicates the raw/modified data for a selected measurement of ~15 min long. Pearson correlation coefficients (PCC) are also shown in the legend as an index of the linear correlation.

the data shows no visible effect on the mean diameters. However, we expect that it would be more influential on the DNC, which will be investigated later in this section.

Since it was already shown in Figure 5 that the two instruments behave differently for small droplets, comparing only the mean diameter values may not be enough for this study. Hence, in Figure 7, we investigated other droplet diameters of the collected data, including the 10th percentile ($D_{10}$), the 50th percentile or median ($D_{50}$), the 90th percentile ($D_{90}$), and the

spectral width, defined as the subtraction of $D_{10}$ from $D_{90}$, showing distribution breadth in each measurement. As in Figure 6, blue and red circles also represent the raw and modified data, respectively, accompanied by the PCC in the legend. Moreover, a second row is added here to demonstrate the results after applying the $D_{min}$ cutoff to the data by filtering out all droplets smaller than the $D_{min}$ value of the PDI from the analysis (same as Figure 5b).

For the majority of the measurements, the percentile values are again higher for the PDI than for the VisiSize, just as for the

mean diameter plots, while in a few cases, they are very close or even on the 1:1 line. The PCCs are also generally close to 1, with the highest linear correlation observed for the largest droplets (90th percentile). In regard to the spectral width, the number of cases very close to or even above the 1:1 line, meaning a larger value for the VisiSize than for the PDI, is higher than in the plots for the diameter percentiles. However, the difference between the raw and modified data is still negligible. By applying the $D_{min}$ cutoff, it can be seen that the scatter plots slightly shift up, especially for the smallest droplets (10th percentile). This

can be a consequence of removing the tiny droplets, which were identified only by the VisiSize, from the analysis. This also affects the spectral width plot and shifts it down slightly. Overestimating droplet sizes by the PDI, as illustrated in Figures 6 and 7, was also observed by Chuang et al. (2008).





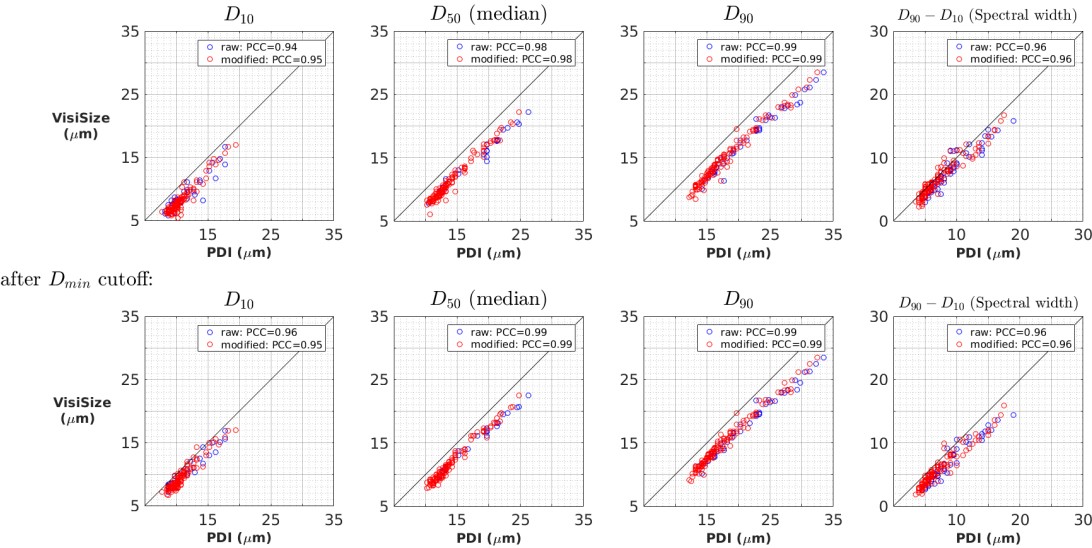

**Figure 7.** Comparison between the VisiSize and the PDI probe regarding the diameter size percentiles ($D_{10}$, $D_{50}$, and $D_{90}$) as well as the spectral width ($D_{90} - D_{10}$) of cloud droplets. Each pair of blue/red circles indicates the raw/modified data for a selected measurement of ~15 min long. The second row shows each plot after applying the $D_{min}$ cutoff to the data. Pearson correlation coefficients (PCC) are also shown in the legend.

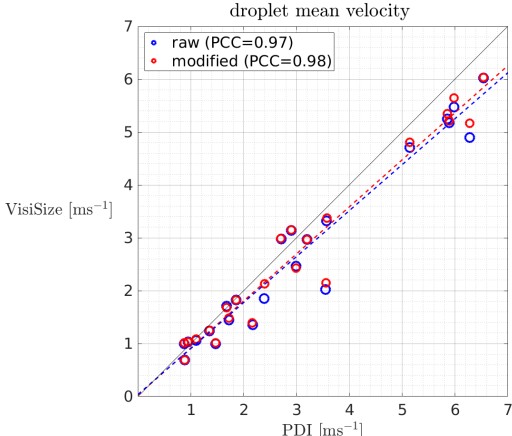

**Figure 8.** Comparison of the mean droplet velocity between the VisiSize and the PDI probe. Each pair of blue/red circles indicates the raw/modified data for a selected measurement of ~15 min long. Dashed lines show the least squares regression (LSR), and the Pearson correlation coefficients (PCC) are displayed in the legend.





### 4.3 Velocimetry

In Figure 8, the cloud mean droplet velocities obtained from the two instruments are compared. As shown in Table 2, there
were only 23 selected measurements with the VisiSize in the velocity mode, where it was able to measure size and velocity
simultaneously. The reason why the VisiSize was mainly used in the diameter mode is bacause the droplet detection rate drops
significantly in the velocity mode, since a single droplet has to be identified twice to be counted. In other words, the device must
find an already detected droplet again in the next frame or the same frame depending on the selected method (see Section 2.1).
Then, it can measure its velocity based on the position shift between the two shadow images. Therefore, the velocity mode has
been used only in the case of dense clouds, which provides enough data for further analysis.

A reasonable correspondence between the mean velocities obtained from the two devices is shown in Figure 8, especially for
smaller velocities. A strong linear correlation between the results also exists based on the PCC values. However, no significant
change is observed after applying the modification to the data. Two additional dashed lines are also plotted corresponding to
each set of data which are "least squares regression" (LSR) lines. It is expected that the closer the LSR line is to the 1:1 line,
the better agreement between the instruments.

Considering the mean velocity as the values increase, one can see a slight change in the trend. The PDI recorded higher mean
velocities than the VisiSize for all cases above $5 \, \text{m} \, \text{s}^{-1}$. A possible explanation could be that in high-speed flows, some droplets
that pass the sample volume very fast can be missed by the VisiSize in the second frame. This occurs due to the limit of the
camera speed (maximum of 30 fps). Hence, the average velocity calculated by the VisiSize would be slightly underestimated
in high-speed flows in comparison with the PDI, which does not experience such a problem.

### 4.4 Droplet Number Concentration (DNC)

Figure 9 shows the comparison between the VisiSize and the PDI probe based on the droplet number concentration (DNC) of
cloud droplets. The first scatter plot on the top-left shows the raw data processed by the instruments, where blue and red circles
represent the original DNC methods for the PDI: transit time and swept volume, respectively. The VisiSize data in this plot
were processed based on the default sample volume (def. SV). Next, in the top-middle, the results after applying the sample
volume correction (cor. SV) to the VisiSize data and using modified DNC methods for the PDI are shown. The top-right plot
indicates the results after applying the coincidence filter to the modified PDI data. Corresponding plots after applying the $D_{\text{min}}$
cutoff are shown in the second row.

The DNC plots are more scattered and demonstrate less linear correlation between the two instrument results than the
mean diameter and velocity plots (Figures $6 - 8$). Therefore, to make a better comparison between the two instruments and
investigate the effects of the applied modifications and filters, the LSR dashed lines and PCC values are also included in the
plots. As mentioned earlier, the VisiSize can fail to detect the fastest droplets in velocity mode, resulting in an underestimation
of the DNC, especially in cloud events with high velocity flows. Thus, for the DNC (and later the LWC) comparison, only the
measurement cases in which the VisiSize was working in the diameter mode were analyzed.





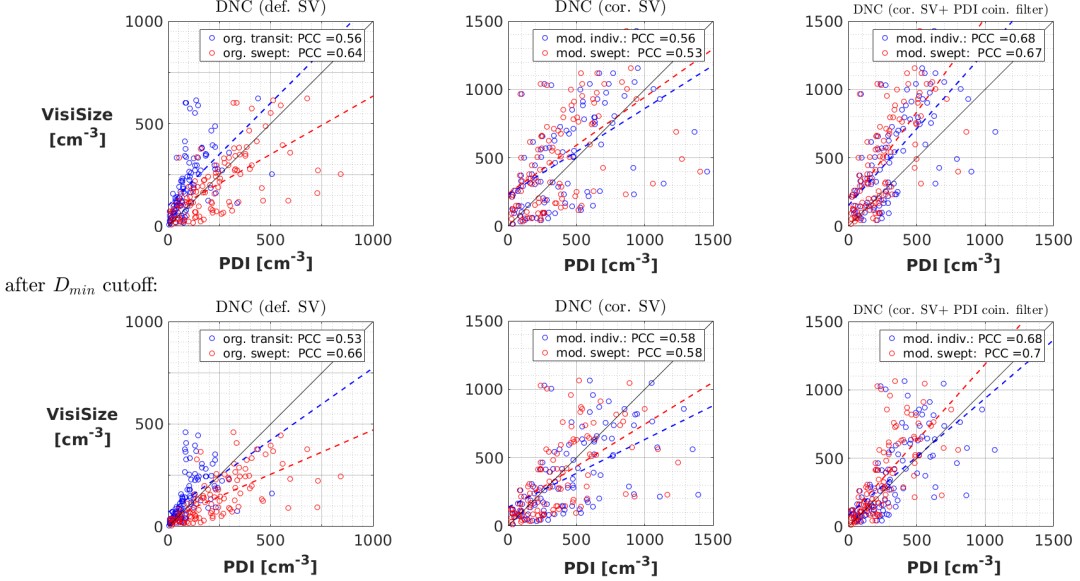

**Figure 9.** Comparison of the DNC results between the VisiSize and the PDI probe. From top-left to top-right, the plots show the raw data (based on the default sample volume for the VisiSize and the original DNC methods for the PDI), the data after modification, and the modified data after applying the coincidence filter to the PDI results. The scatter plots after applying the $D_{min}$ cutoff are shown in the second row. Each PDI method for the DNC computation is shown with a different color within the plots (each pair of blue/red circles: a selected measurement of ~15 min long). Dashed lines show the least squares regression (LSR), and the Pearson correlation coefficients (PCC) are displayed in the legend.

The first plot in Figure 9 (top-left) shows a noticeable discrepancy between the transit time and the swept volume methods. The DNC values from the transit time method of the PDI are usually smaller than the VisiSize raw data, as the LSR line has an ~100 cm$^{-3}$ shift from the 1:1 line toward the VisiSize. However, with the swept volume method, there were more cases with higher DNCs for the PDI than for the VisiSize. Subsequently, after applying the SV correction on the VisiSize data as well as modification of the PDI swept volume method in the next plot (top-middle), the DNC values generally increase. Nevertheless,

in most cases, especially in dense cloud events (high DNCs), the VisiSize provides higher values than the PDI. The shift of the results toward the VisiSize continues after applying the coincidence filter to the PDI data. Finally, after omitting the small droplets by means of the $D_{min}$ cutoff, the LSR lines tend toward the 1:1 line, especially for the individual method of the PDI, with a slight growth in the PCC values to their highest level. This behavior after applying the $D_{min}$ cutoff agrees well with what has already been observed in Figure 5.

In Figure 9, we observed that after excluding the smallest droplets, the correspondence between the two instruments regarding the DNC results improved. Therefore, it is important to investigate the instrument responses to droplet size change in more detail in regard to the DNC computation. For this purpose, the DNC ratios of the VisiSize to the PVC-PDI as a function of arithmetic mean diameter size are plotted in Figure 10. In the left panel, the raw data from the instruments are compared. The





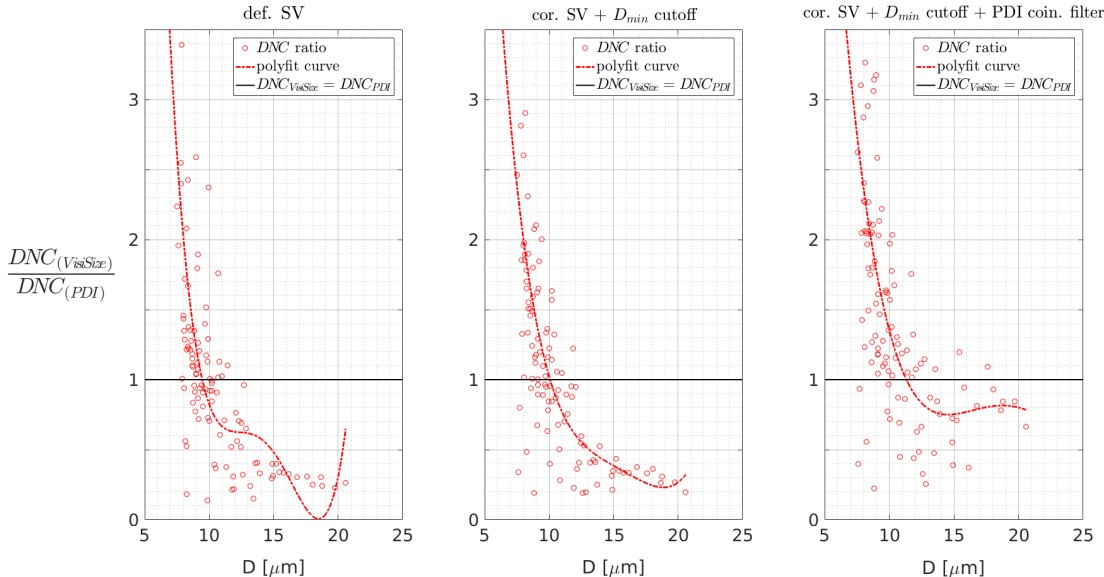

**Figure 10.** The DNC ratio of the VisiSize to the PVC-PDI as a function of the arithmetic mean diameter size. Panels from left to right show the raw data (using the default sample volume for the VisiSize), the results after the SV correction and the $D_\mathrm{min}$ cutoff, and the results after additionally applying the coincidence filter to the PDI data. Each circle represents a single measurement of ∼15 min long. The polynomial fit curves and the 1:1 ratio are shown with the dashed and black horizontal lines, respectively.

swept volume method from the PDI with consideration of the PVC factor is used and called PVC-PDI. Just as in previous plots,
each circle represents a single selected measurement, while a polynomial fit curve is also plotted to better illustrate the trend as the arithmetic mean diameter increases. The mean droplet size for each measurement is taken from the VisiSize raw data. The 1:1 ratio lines where the DNC results from the two instruments are equal are also shown within the plots. Plots after applying modifications and filters to the data are shown in the next panels in the middle and right side of Figure 10.

For the smallest sizes, the ratio is higher than 1, which means that the raw values from the VisiSize are actually a couple
of times larger than those from the PDI. However, as the mean size increases, there is a dramatic decrease, and for mean diameters of ∼10 μm, there are many cases close to the 1:1 ratio line. As the mean size increases, the ratio decreases more, and for $\bar{D} > 13$ μm, it even reaches below 0.5. Nevertheless, we observe gradual changes after applying the modifications, $D_\mathrm{min}$ cutoff, and finally, the coincidence filter, which makes the DNCs show better correspondence, especially for larger droplets.

The effect on the DNC results of using different camera lens magnification settings in the VisiSize measurements is inves-
tigated in Figure 11. The last plot from Figure 9, where the results were shown after applying all modifications and filters to the data, is replotted here for measurement cases with each VisiSize lens magnification separately. In the left panel, the PVC-PDI results from the modified swept volume method are compared with the VisiSize results with a magnification of ×2. The comparison of the measurements with a magnification of ×4 is shown in the right panel. In the case of using a magnification of ×2, the results are still generally scattered, and the LSR line has a ∼100 cm⁻³ shift toward the VisiSize from the 1:1 ratio



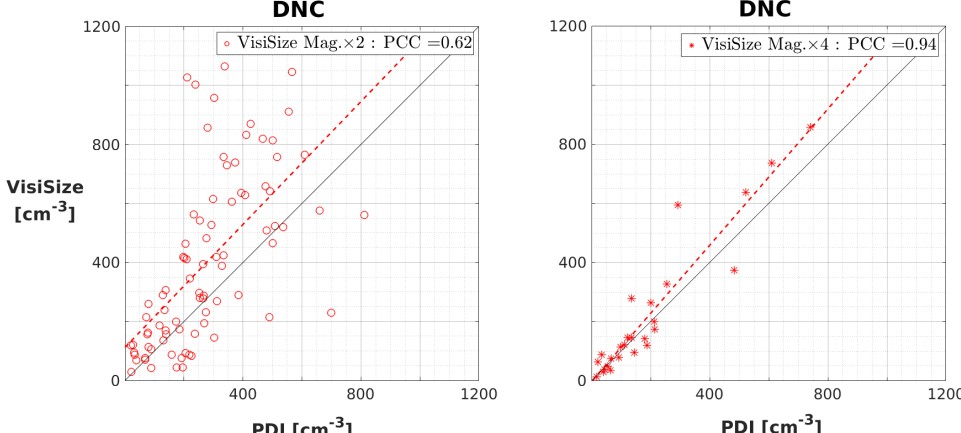

**Figure 11.** The DNC comparison between the two instruments, with cases using different lens magnification settings for the VisiSize ($\times 2$ and $\times 4$), plotted separately. The results after applying all modifications and filters to the data and application of the PVC factor to the modified PDI swept volume method are shown here. Each circle/star represents a single measurement of $\sim 15$ min long. Dashed lines show the least squares regression (LSR), and the Pearson correlation coefficient (PCC) is shown in the legend.

line. Nonetheless, the results of using a magnification of $\times 4$ interestingly show better agreement between the two devices. The PCC value is larger, and the LSR line is closer to the 1:1 ratio line, especially for smaller DNCs. However, the number of cases measured with $\times 4$ magnification is approximately three times lower than those measured with $\times 2$ magnification (see Table 2). As mentioned earlier, for higher lens magnifications of the VisiSize, despite the rise in sizing accuracy, the sample volume (SV) decreases significantly. Therefore, it is sometimes difficult to collect a sufficiently large sample during the course

of measurements, especially for larger droplets. Thus, we often used the magnification of $\times 2$ to collect sufficient data in each measurement, while the use of the $\times 4$ magnification was limited mostly to the events with smaller cloud droplets.

In Figure 12, the DNC results from the VisiSize and the PDI probe are again shown after applying all modifications and filters. The diameter arithmetic mean ($\bar{D}$) and the velocity standard deviation ($\sigma_v$) within the measurement cases are also illustrated by adding a color map next to the scatter plot. The $\bar{D}$ was calculated using the VisiSize raw data, while the PDI raw

data were used for the $\sigma_v$. As in Figure 10, for smaller droplets here, the DNCs from the VisiSize are higher than those from the PDI, while with a rise in $\bar{D}$, they correspond better. This is also in agreement with the Chuang et al. (2008) observation, where the PDI reported a lower DNC than the FSSP-100 device, especially for smaller size ranges. On the other hand, the DNC response with respect to the changes in velocity does not seem to follow the same trend. The DNCs in cases with the lowest $\sigma_v$ mostly agree better. This was expected since the PDI requires averaging droplet velocity over the size/time bins to compute the

DNC, which can be a source of inaccuracy (see Eqs. 2 and 3). However, there are also cases with low a $\sigma_v$, where the results do not correspond well. Hence, it is difficult to draw a solid conclusion regarding the effect of the $\sigma_v$ on the DNC response.





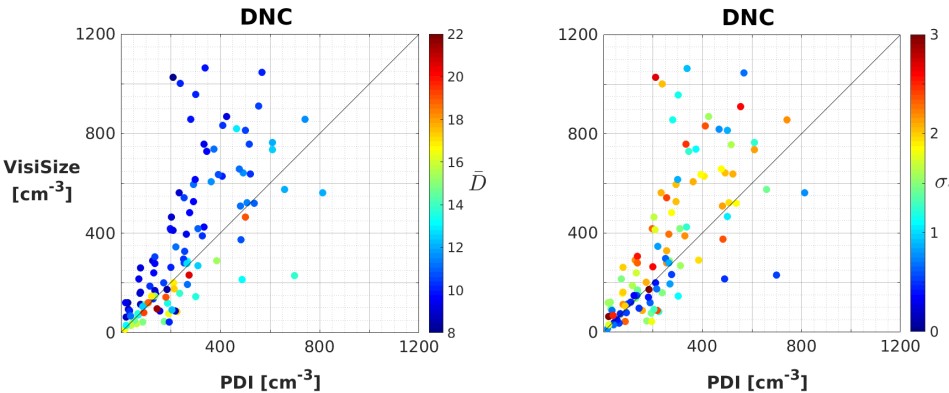

**Figure 12.** The DNC comparison between the two instruments, with the mean diameter spectra (left panel) and the standard deviation of velocity spectra (right panel) compared between the selected cases. The results after applying all modifications and filters to the data and with application of the PVC factor to the modified PDI swept volume method are used here. The mean diameters and the velocity standard deviations are calculated using the VisiSize and PDI data, respectively. Each circle represents a single measurement of ∼15 min long.

## 4.5 Liquid Water Content (LWC)

In Figure 13, the same scatter plot pattern as in Figure 9 is used to demonstrate the LWC comparison between the two instru-ments. Although both PDI methods report a higher LWC than the VisiSize, the discrepancy between the swept volume and the

VisiSize is greater. The SV correction alone does not help reduce this discrepancy, yet it increases the LWC values. However, after applying the coincidence filter, the LSR line slope increases and moves toward the 1:1 ratio line slightly. Nonetheless, after applying the $D_{\min}$ cutoff, the changes are negligible.

Explaining why discrepancies in the LWC are generally larger than in the DNC, while the applied filters are less effective in reducing them, requires considering the main variables related to the LWC. The mass of the water droplets is one of these

variables, which can be replaced by the droplet diameter size, assuming constant water density and spherical droplet shape. The other key component of LWC computation is the DNC itself. Therefore, changes in the LWC are directly related to the DNC as well as the volume mean diameter ($D_3$). In other words, the discrepancies in both the DNC and mean droplet size facilitate a greater resultant discrepancy in the LWC. However, it is crucial to bear in mind that while the LWC changes linearly with the DNC, it increases cubically with a rise in the mean diameter size (LWC $\propto$ DNC $\cdot D_3{}^3$). Consequently, smaller droplets play a

less important role in the LWC estimates than larger droplets. Then, excluding the smaller droplets by applying the $D_{\min}$ cutoff would also have less effect on the LWC than on the DNC.

In Figure 14, the LWCs from the two instruments are compared for different VisiSize lens magnifications, as shown in Figure 11. There is no significant difference between the magnifications regarding the LWC results, and in both, the LWC leans toward




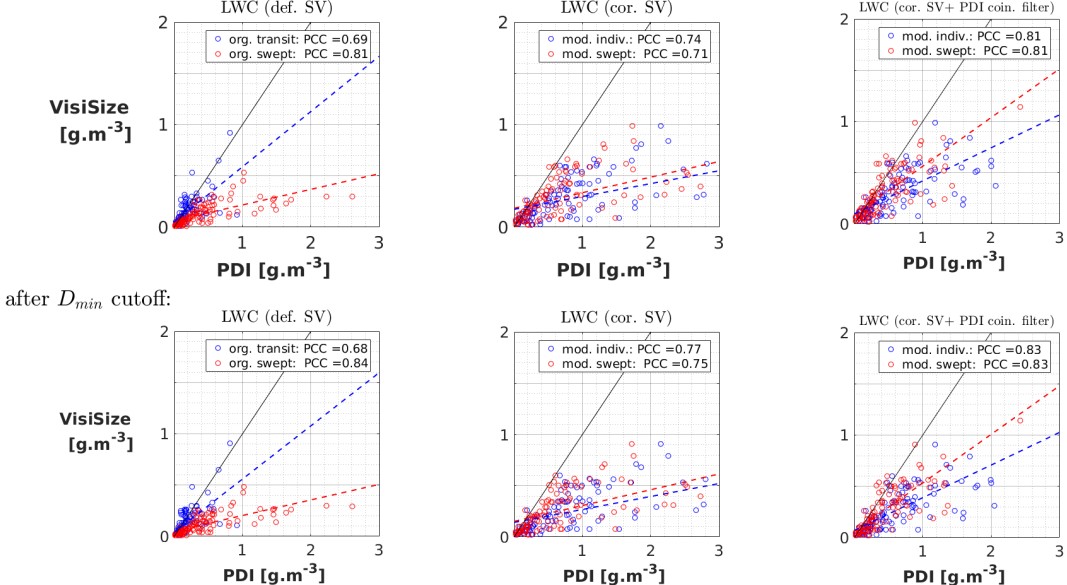

**Figure 13.** Comparison of the LWC results between the VisiSize and the PDI probe. From top-left to top-right, the plots show the raw data (based on the default sample volume for the VisiSize and the original DNC methods for the PDI), the data after modification, and the modified data after applying the coincidence filter to the PDI results. The scatter plots after applying the $D_{min}$ cutoff are shown in the second row. Each PDI method for the LWC computation is shown with a different color within the plots (each pair of blue/red circles: a selected measurement of ∼15 min long). Dashed lines show the least squares regression (LSR), and the Pearson correlation coefficients (PCC) are displayed in the legend.

the PDI, as in Figure 13. Subsequently, the comparison between the LWCs is shown in Figure 15, while the changes in the mean diameter value and the velocity standard deviation are also illustrated with the color maps next to the plots. As shown in the left panel, the two instruments show better agreement for smaller droplet mean values, while for larger droplets the discrepancy is higher.

Although the trend here is opposite to what was observed for the DNC, it is still explicable as we consider the cubical relation between the LWC and the mean droplet size. Thus, the LWC discrepancy arises as the mean diameter size increases.
On the other hand, for small droplets, the lower discrepancy in the mean size, where the PDI shows larger values (shown in Figures 6 and 7), is cancelled here by the discrepancy in the DNC, where the VisiSize values are higher (see Figure 12). In addition, it is difficult to find a clear trend by observing the right panel in Figure 15 where the velocity standard deviation changes in the LWC scatter plot are shown. However, it seems that contrary to Figure 12, cases with a lower $\sigma_v$ do not represent better correspondence between the two devices. The PDI mostly shows higher LWC values for these cases. This can also be
because the LWC is more influenced by the discrepancy in the mean droplet sizes than the DNC.



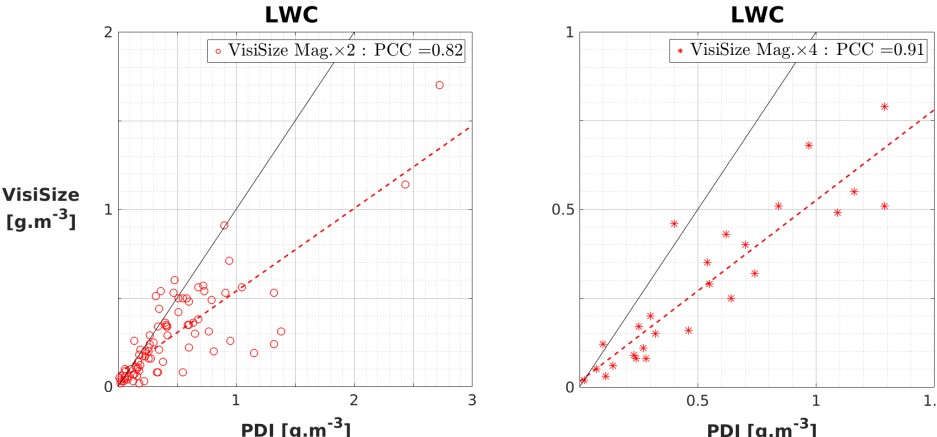

**Figure 14.** The LWC comparison between the two instruments, with cases using different lens magnification settings for the VisiSize ($\times 2$ and $\times 4$), plotted separately. The results after applying all modifications and filters to the data and application of the PVC factor to the modified swept volume method of the PDI are shown here. Each circle/star represents a single measurement of $\sim 15$ min long. Dashed lines show the least squares regression (LSR), and the Pearson correlation coefficient (PCC) is shown in the legend.

## 5 Conclusions

Cloud droplet microphysical properties, such as mean droplet diameters ($\bar{D}, D_2, D_3$), droplet size distribution (DSD), droplet number concentration (DNC), and liquid water content (LWC), were measured. The measurements took place during a 3-week campaign in July and August 2019 at the UFS, a mountain observatory located below the peak of Mount Zugspitze in the German Alps. After successful laboratory experiments (Nowak et al., 2021), the VisiSize D30, a commercial shadowgraph instrument manufactured by Oxford Laser Ltd., was utilized for the first time for sizing and velocimetry of cloud droplets. Meanwhile, a PDI-FPDR probe manufactured by Artium Tech. Inc. was also simultaneously measuring the cloud droplet microphysical properties at the UFS.

During the course of measurements, 133 cases were selected for further analysis. Each of those selected cases was approximately 15 minutes long. They involved different VisiSize lens magnification settings ($\times 2$ and $\times 4$) as well as running modes. By default, correction of the size-dependent sample volume (SV) of the VisiSize was performed based on Nowak et al. (2021). Moreover, the DNC computation method of the PDI was also modified to reduce the effect of velocity averaging. Subsequently, the preliminary analysis of the data led us to apply an additional coincidence filter and then a cutoff on the minimum droplet size so that very small droplets were excluded from the analysis to make a better comparison between the two instruments.

The DSD comparison revealed that the two instruments perform differently for the measurement of small droplets, while for larger droplets ($D > 13$ µm), the results correspond better. Generally, the VisiSize displayed higher sensitivity than the PDI probe in the detection of very small droplets, although it has a slower detection rate since it works based on a different optical





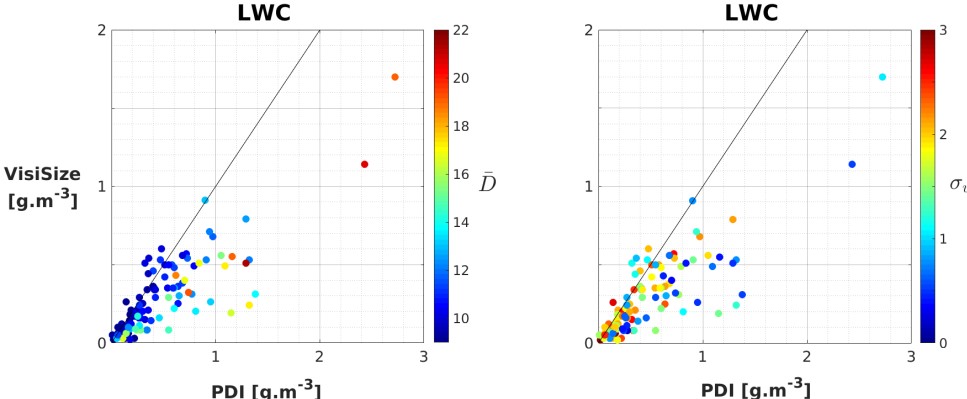

**Figure 15.** The LWC comparison between the two instruments, while the mean diameter spectra (left panel) and the standard deviation of velocity spectra (right panel) are also compared between the selected cases. The results after applying all modifications and filters to the data, and with application of the PVC factor on the modified swept volume method of the PDI are shown here. The mean diameters and the velocity standard deviations are taken from the VisiSize and the PDI data, respectively. Each circle represents a single measurement of ∼15 min long.

principle. Consequently, the mean droplet diameter results from the PDI were consistently between 3 and 6 μm higher than the results from the VisiSize. This was also observed by Chuang et al. (2008), where the PDI was compared with a FSSP-100.

Regarding the mean velocity of the droplets, the two instruments show better agreement, especially in the cases with lower mean velocity, while in high-speed flows, the VisiSize seems to underestimate a little of the mean velocity due to the camera speed limit.

A comparison between the DNC results also showed a discrepancy between the two instruments, similar to the Chuang et al. (2008) observations. We applied a SV correction to the VisiSize data, modifications to DNC methods of the PDI, coincidence

filtering to the PDI data, and a $D_{\min}$ cutoff to the data of both instruments. Subsequently, the effects of droplet size, velocity fluctuation, and lens magnification of the VisiSize on the DNC were investigated. As a result, the discrepancy between the DNCs was generally attributed to the different performances regarding smaller droplets, as well as the constraint of the PDI on velocity averaging for the DNC measurments (see Eqs. 2 and 3). The latter increases the uncertainty when experiencing highly turbulent clouds at the UFS. The PDI probe was mainly designed for airborne measurements, where the velocity fluctuations

are negligible compared to the mean flow velocity. Moreover, the LWC measurements were also compared, and the discrepancy was even greater than the DNC, and the filters were less effective. This can be because both discrepancies in the mean droplet size and the DNC play a role in the LWC results.





In conclusion, the VisiSize D30 shadowgraph instrument has been successfully applied to cloud microphysics measurements. Furthermore, comparing the VisiSize with the PDI probe after applying the SV correction and filters to data led to a better

correspondence in sizing and velocimetry than the DNC or the LWC measurements. There are still difficulties in computing an accurate SV for a given cloud droplet size, especially for the PDI probe, causing discrepancies in the DNC and LWC results.

*Code availability.*   The results presented in this study were obtained with the use of the Oxford Lasers VisiSize software version 6.5.39 and code developed by the authors in the MATLAB environment. The latter are available from the authors upon request.

*Data availability.*   The data presented in this study are available from the authors upon request.

**Appendix A:  PDI channel 1 data**

As mentioned in Section 3, the PDI channel 1 did not function properly during the summer 2019 campaign. Hence, all PDI results presented in Section 4 were obtained from channel 2, which is supposed to be less sensitive in the detection of very small droplets. However, there were some data available from measurements at Mount Zugspitze in summer 2018, where the PDI channel 1 functioned reliably in cloud droplet measurements. These experiments were conducted with both the VisiSize

and PDI channel 1 at the same measurement site at the UFS. Nonetheless, the number of simultaneous measurements with large enough collected samples was so limited that performing comprehensive analysis on them was difficult. In addition, the available simultaneous measurements also have different lengths varying between $\sim 7-14$ minutes. Thus, we could select only 12 measurement cases to show a comparison between the VisiSize and PDI channel 1 regarding the arithmetic mean diameter of droplets in Figure A1.

The result is the same as that shown in Section 4 for the PDI channel 2. The arithmetic mean from the PDI channel 1 is consistently higher than that from the VisiSize, just as it was higher from the PDI channel 2 (see Figure 6). However, the difference in the mean values from the two instruments is lower when we used the VisiSize lens magnification of $\times 2$ than cases measured with $\times 4$ magnification. The mean values from the PDI and VisiSize with $\times 2$ magnification differ between 2 and 3 μm, while the PDI camparison to the cases with $\times 4$ magnification shows between 2 and 6 μm difference. As expected,

the higher lens magnification in the VisiSize is more sensitive in the detection of the smallest cloud droplets which results in a lower mean value and consequently a larger difference from the PDI results. Therefore, the two instruments correspond better in sizing cloud droplets when we use the VisiSize lens magnification of $\times 2$ and the PDI channel 1.

*Author contributions.*   SPM conceptualize the research. MM, GB, JM and WK performed the measurements during the campaign. MM and JLN analyzed and interpreted the collected measurement data. MM wrote the original manuscript with contributions from JLN, GB, JM and
SPM.



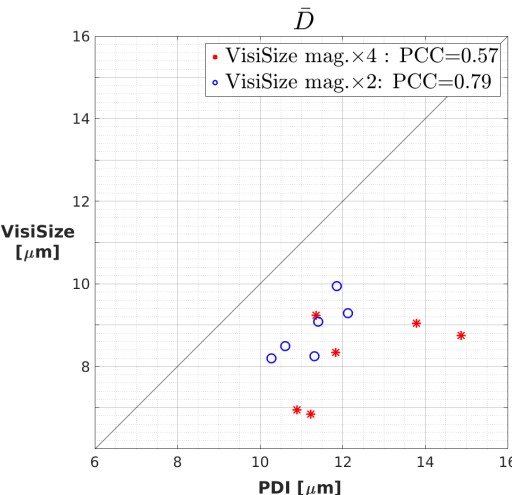

**Figure A1.** Comparison of the arithmetic mean droplet diameters between the VisiSize and the PDI probe channel 1. The blue circle represents measurement cases where the VisiSize lens magnification of ×2 was used, while the red star shows the measurements with ×4 magnification. Each measurement case is between ∼ 7 − 14 minutes long, while both instruments were collecting cloud droplet data simultaneously. The Pearson correlation coefficient (PCC) is also shown in the legend.

*Competing interests.* The authors declare that they have no conflict of interest.

*Acknowledgements.* We are grateful to Dr. Till Rehm and the staff of the Umweltforschungsstation Schneefernerhaus (UFS) for their help during the field measurements at Mount Zugspitze.



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
