# Peer review of "Cloud microphysical measurements at a mountain observatory: comparison between shadowgraph imaging and phase Doppler interferometry"

_Atmospheric Measurement Techniques, 2021_

## Author Comment (AC1)

**Authors' response to the anonymous referee #2**

Moein Mohammadi, Jakub L. Nowak, Guus Bertens, Jan Moláček, Wojciech Kumala, and Szymon P. Malinowski

We are grateful to the referee for the insightful comments and suggestions on our manuscript. We respond to them in detail below. The original review is given in black, and our answers in blue. The responses mention also specific corrections which were applied to the manuscript.

1. Figures 3 and 4 and the associated text leave some confusion about the orientation of the instruments relative to the wind direction. On line 225 I understand that the instruments are facing in a direction that allow optimal sampling of wind coming from 320 degrees. This could be indicated in the last panel of Figure 4, or at least mentioned in the caption. If that understanding is correct, what does it imply about the measurements on 13-Jul, 14-Jul, and 13-Aug when the wind direction is close to 90 degrees? Is it expected that the measurements are heavily influenced by the instrument housings? That may cause problems with the sample volume estimation for the phase Doppler interferometer. This is relevant to the presented results, e.g., see line 263 where average wind direction is stated as 225 degrees. Please address this aspect of the study, and clarify the directions as needed.

   Thanks to the referee's comment, we realised that the information in line 225 regarding the instrument orientation is misleading. Hence, in the revised manuscript we have modified that part. In fact, the main parts of the VisiSize (laser head and camera) as well as the PDI (transmitter and receiver) were aligned in almost north-south direction (with at most $10°$ deviation to the west) as shown in Fig. 1 below. This allows optimal sampling of clouds coming from a perpendicular direction (with westerly winds). A mountain ridge named "Schneefernerkopf" located in west of the UFS has a part eroded over 200 m long, known as "wind hole" which directs the westerly wind over the UFS like a funnel (Risius et al., 2015). As mentioned in the manuscript and shown in Fig. 4 there, the average wind direction over the course of measurements was $\sim 225°$ with most cases between $180°$ and $270°$. Hence, thanks to the setup configuration schematically shown here in Fig. 1, we were able to optimally measure the clouds which in most cases were coming from the wind hole. Nevertheless, the above explanation has been also added to the revised manuscript in Section 3 to clear up the confusion over the orientation of the instruments.

   With respect to the three days with mostly easterly winds (with direction $\sim 90°$) that is mentioned in the referee's comment, the same data analyses as in Section 4 were repeated after excluding cases with easterly winds. However, the changes in the final results were negligible. For instance, Fig. 2 here shows the DNC comparison between the two instruments (same as the last panel of Fig. 9 in manuscript) with and without cases of wind from east. Comparing the PCC values and LSR lines slopes between the two plots shows almost the same results as mentioned above.

[Figure]

**Fig. 1** The schematic representation of the experimental setup orientation during the Zugspitze measurement campaign at the UFS. This configuration allows to optimally measure the clouds coming with the wind from the "wind hole" located on the Schneefernerkopf in west of the UFS.

In general, we did not expect that easterly wind would affect sizing or velocimetry measurements with the VisiSize since its laser head and camera housing are off-axis. However, the velocities measured with the PDI could be slightly affected in case of easterly winds due to the probe's shape. Nevertheless, we believe that those cases (with easterly wind) did not occur often, and then if occurred the wind speed was quite low ($\sim$1 m.s$^{-1}$), which could explain why their exclusion from the analyses did not change the final results. When the wind speed is low, the wake area behind the body of the PDI is expected to be relatively small as well. Hence, the instrument's sample volume is rather weakly affected by the wake area. The observation of weaker easterly winds compared to westerly winds has been also mentioned by Risius et al. (2015) (Fig. 4 there).

The above mentioned point has been addressed within the revised manuscript at the end of Section 3.

[Figure]

**Fig. 2** Comparison of the DNC results between the VisiSize and the PDI probe with and without data collected during easterly wind events. The modified data after applying the coincidence filter to the PDI results as well as $D_{min}$ cutoff on both instruments are shown here.

2. In the presentation of results on droplet number concentration (e.g., Figure 9) please provide some information on the range of number of droplets sampled for the data points that are shown. That would allow Poisson uncertainty to be evaluated. I expect the numbers are large enough that Poisson sampling uncertainty is not a significant contribution, but please show or discuss this directly.

The information regarding the typical range of droplet counts as well as Poisson sampling uncertainty during the measurements has been included in the revised manuscript as follows:

"Typical range of droplet counts during the measurements was between $560-10463$ with the VisiSize and between $6330-118330$ with the PDI. Subsequently, the Poisson sampling uncertainties for the average (arithmetic mean) count with the VisiSize and the PDI are estimated to be $\sim1.3$ and $\sim0.4\%$ respectively".

3. In several of the plots I believe more insight could be obtained by showing results on logarithmic axes. For example, log-log coordinates might help in the interpretation of the droplet number concentration results. But for sure it would be helpful in the presentation and interpretation of the LWC results in Figure 13. This would allow a broader range of LWC to be observed, rather than results being compressed near the origin because of a relatively small number of outlier points. I expect the least-square fits will be significantly different in log-log coordinates as well.

The authors entirely concur with the referee's comment. Therefore, all plots in the last two subsections (4.4 and 4.5), including the results of the DNC and LWC analyses, have been transformed into log-log coordinates. Subsequently,

the least square regression (LSR) lines and values of Pearson correlation coefficients (PCC) have been also updated. Moreover, the explanation regarding each figure within the manuscript has been modified accordingly, if needed. Indeed, using the logarithmic scale improved the quality of the resulting presentation.

4. Another comment on data presentation: in Figure 10 it would make sense to show the y-axis in log scale so that the full variability can be observed. The polynomial fit seems rather arbitrary. Is there any justification for this form? Perhaps log-scale will suggest something else.

Following the referee's suggestion in the previous comment, the vertical axes in Fig. 10 have been shown in logarithmic scale (see Fig. 3 here). The polynomial fit was selected first as one of the most simple functions merely to show the downward trend of the DNC ratio as the mean droplet size increases. Subsequently, after transforming the y coordinate into logarithmic scale, we changed the fit curve into reciprocal ($y=\frac{a}{x+b}$) which performs better for this purpose in logarithmic scale.

[Figure]

**Fig. 3** The DNC ratio of the VisiSize to the PVC-PDI as a function of the arithmetic mean diameter size. Panels from left to right show the raw data (using the default sample volume for the VisiSize), the results after the SV correction and the $D_{min}$ cutoff, and the results after additionally applying the coincidence filter to the PDI data. Each circle represents a single measurement of ~15 min long. The reciprocal fit curves ($y = \frac{a}{x+b}$) and the 1:1 ratio are shown with the dashed and black horizontal lines, respectively.

5. Another comment related to Figure 10: At large mean diameter there was a strong undercounting of the shadowgraph method or overcounting of the PDI method. If you use the polynomial fit to 'adjust' for this bias, how much improvement would result in the LWC results in Figure 13? This would be one way to further disentangle the number concentration

and $D^3$ contributions. It seems possible that the PDI instrument is overestimating number concentration when the mean size is large, and that this is leading to the unreasonably high values of LWC.

Following the referee's suggestion, we used the fitted values from Fig.10 in order to adjust the DNC from the PDI which is used in the LWC calculation shown in Fig. 13. Consequently, the comparison between the PDI and the VisiSize liquid water content after applying modifications and filters on data is shown in Fig. 4 below.

[Figure]

**Fig. 4** Comparison of the LWC results between the VisiSize and the PDI probe after adjusting the DNC data of the PDI with a reciprocal fit to the DNCs resulted from the VisiSize data. The scatter plot is shown after correction of the SV, applying the coincidence filter to the PDI results, and D$_{min}$ cutoff to both instruments' data.

Comparing this plot with the last panel of Fig. 13 in the manuscript, one can see the improvement in the correlation between the data from the two instruments. The PCC values increased by 12% (from 0.84 to 0.94) after adjusting the DNC values from the PDI, using reciprocal fit in Fig. 10. Moreover, the LSR lines are also a little closer to 1:1 line here than in Fig. 13, which is expected as the PDI data were adjusted according to the VisiSize data. However, the LWC is related to both the DNC and the volume mean diameter (LWC $\propto$ DNC $\cdot D_3{}^3$). Therefore, due to the difference in D$_3$ measured by the two instruments, there is still discrepancy in the LWC results.

The Fig. 4 has been added to the revised manuscript along with the above explanation.

6. The approach of the study is an instrument intercomparison, but without a known reference or standard. Generally, the authors point out differences without suggesting that any specific technique is correct or incorrect, and I agree with that cautious approach. (Note that one exception is on line 311, which should be reworded, or evidence shown that

one method is indeed overestimating size.) In some places, however, it may be possible to argue that one instrument is more closer to reality, such as in Figure 13 where the PDI is showing significant numbers of data points with liquid water contents above 1 g.m$^{-3}$. Based on other measurements at the UFS and similar mountain stations, I expect these values are much too high. More importantly, the paper is missing a discussion in the Conclusions section about how the investigation could be taken to the next level. Is there a way to perform an absolute calibration or provide additional instruments in future studies, so that the discrepancies can be understood and resolved? In the end, the community needs to know which of these instruments can be considered reliable, in what range of conditions.

The line 311 has been reworded to be more clear. The argument regarding the LWC results has been also added to the revised manuscript with reference to Siebert et al. (2015) and Spiegel et al. (2012). In these studies, the LWC has been measured with PMV-100A (Gerber et al., 1994) during series of ground-based measurements on Alps (and in case of Siebert et al. (2015) at the UFS). The typical values measured for the LWC in both studies were below 1 g.m$^{-3}$, which suggests that the PDI in our measurements overestimated the LWC, especially in case of clouds with larger mean droplet sizes.

To conduct further investigations, laboratory tests with standard glass beads or monodisperse droplet generator can be useful to compare the sizing accuracy between the instruments. Moreover, the instruments can be compared with a particle tracking experiment in the laboratory, which would be mostly beneficial for evaluating the DNC results. However, it should be noted that even under laboratory conditions, simultaneously controlling size, velocity, and number concentration of samples would be challenging. In conclusion, we believe it is essential to work on improving the sensors in order to remove the sizing bias and to enhance the sample volume estimation accuracy.

The above explanations have been also added to the conclusion section of the revised manuscript.

**References**

Gerber, H., Arends, B. G., and Ackerman, A. S.: New microphysics sensor for aircraft use, Atmospheric Research, https://doi.org/10.1016/0169-8095(94)90001-9, 1994.

Risius, S., Xu, H., Di Lorenzo, F., Xi, H., Siebert, H., Shaw, R. A., and Bodenschatz, E.: Schneefernerhaus as a mountain research station for clouds and turbulence, Atmospheric Measurement Techniques, https://doi.org/10.5194/amt-8-3209-2015, 2015.

Siebert, H., Shaw, R. A., Ditas, J., Schmeissner, T., Malinowski, S. P., Bodenschatz, E., and Xu, H.: High-resolution measurement of cloud microphysics and turbulence at a mountaintop station, Atmospheric Measurement Techniques, https://doi.org/10.5194/amt-8-3219-2015, 2015.

Spiegel, J. K., Zieger, P., Bukowiecki, N., Hammer, E., Weingartner, E., and Eugster, W.: Evaluating the capabilities and uncertainties of droplet measurements for the fog droplet spectrometer (FM-100), Atmospheric Measurement Techniques, 5, 2237–2260, https://doi.org/10.5194/amt-5-2237-2012, https://amt.copernicus.org/articles/5/2237/2012/, 2012.

---

## Author Comment (AC2)

**Authors' response to the anonymous referee #1**

Moein Mohammadi, Jakub L. Nowak, Guus Bertens, Jan Moláček, Wojciech Kumala, and Szymon P. Malinowski

We are grateful to the referee for the insightful comments and suggestions on our manuscript. We respond to them in detail below. The original review is given in black, and the answers in blue. The responses mention also specific corrections which were applied to the manuscript.

1. In the introduction of the coincidence filter, it would be good to explain why this effect happens. If I understand correctly, you assume that there was only one droplet, but it got somehow recorded twice by the instrument and you only keep the average size and velocity. Could actual coincidence also be a contributing factor, if two droplets (or fragments) pass the sample volume? Since other instruments have issues with recording two drops at a time, it would be worth noting.

   Typically in instruments which count droplets one-by-one, a measurement artifact known as "coincidence" occurs when more than one droplet are registered at the same time resulting in multiple droplets artificially measured as one droplet. With a rise in droplet number concentration, the coincidence also increases. It has been extensively studied for Forward Scattering Spectrometer Probe (FSSP), Cloud Droplet Probe (CDP), and Cloud and Aerosol Spectrometer (CAS) (e.g. Cooper, 1988; Burnet and Brenguier, 2002; Lance et al., 2010; Lance, 2012). An optical modification was applied to the CDP by Lance (2012) in order to reduce over-sizing and under-counting biases due to coincidence.

   On the other hand, for the PDI as the probe volume, which is defined by the intersection of two laser beams, can be easily reduced, the occurrence of coincidence is expected to be less likely. The probe volume diameter of the PDI, which is determined by the beam diameter, laser wavelength, and transmitter focal length, has been optimized by the manufacturer to minimize the coincidence error. For instance, the probe volume for a typical cloud droplet size of 15 $\mu$m is estimated to be smaller than $10^{-5}$ cm$^3$. Considering a dense cloud with a droplet number concentration equal to 1000 cm$^{-3}$, the average number of droplets within the probe volume would be less than 0.005. Hence, assuming a homogeneous droplet distribution (i.e. disregarding droplet clustering), the probability of having more than one droplet in probe volume, given by a Poisson distribution, would be less than 0.000012. Consequently, the occurrence of coincidence is presumably unlikely.

   Nevertheless, the effects of possible coincidence on PDI data have been discussed in Chuang et al. (2008). If the coincident droplets have similar sizes it is quite likely that both would be rejected by the PDI since the Doppler burst and phase differences in this case would be different than what should appear for a single droplet. Conversely, if the difference in sizes of coincident droplets would be high, the signal from the larger one would be usually dominant and it would be

detected, while the smaller droplet is missed. Hence, the under-counting of the small and mid-sized droplets in cloud can be the consequence of coincidence, while the large droplets are counted accurately by the PDI.

During our measurements at the UFS, however, we faced coincidence phenomenon in a slightly different way than mentioned above. In raw data of cloud droplets collected by the PDI, simultaneously recorded counts were found with identical detection times. Their size and velocity values were similar with differences usually less than 2.0 $\mu$m and 0.1 m.s$^{-1}$, respectively. Since these coincident counts with similar sizes were recorded instead of being rejected, we assumed they are actually one droplet recorded more than once by the PDI, which is considered a consequence of multiple triggering in Chuang et al. (2008) research. This error was attributed to the noisy environment by Chuang et al. (2008) which in our case could be the droplets flow passing between the transmitter and probe volume resulting in fluctuations of laser light intensity at the beam intersection. In order to reduce its effect on the results, we have used a coincidence filter in which all simultaneously recorded counts are averaged in size and velocity. It should be also noted that the coincidence filter does not correct the data for the improbable events of multiple droplets simultaneously present in the PDI probe volume.

Following the referee's suggestion, the above mentioned explanation has been also included to the manuscript in Section 2.2.3.

2. For Figure 5, a size distribution of dNdlogd would be more useful. This way, it would be easy to see at which sizes the two instruments deviate. Since they seem to match in panel d, a dNdlogd size distribution would also show agreement there and show a deviation for smaller sizes. This would also show a comparison in the number concentration. Right now, it is not easy to see from panel a if the PDFs should be shifted or if one instrument detects less particles than the other; that only becomes obvious when considering panel d. This way, you could also include both methods for the PDI to show if those two have the same size distribution but shifted up/down (due to the different sample volume) or if they are skewed. It would help to also understand table 3. Probably the size distributions in Figure 5 only match for very large cutoff because $D_{min}$ should not be a sharp cutoff of the PDI, but it might miss smaller droplets gradually?

The authors generally concur that providing size distribution of dN/dlogD is useful as it combines the information regarding both droplet size distribution (DSD) and droplet number concentration (DNC) in one plot. However, at that point in the manuscript, we focused more on the comparison of cloud droplet sizing properties between the two instruments. Using the probability distribution function (PDF) allows us to merely compare sizing performances between the VisiSize and the PDI. The inaccuracy in estimation of the sample volumes of the instruments which directly affects the DNC (and the LWC) results does not influence the PDFs. Hence, this comparison would be independent of the estimation of sample volumes whose effects on the DNC and LWC results has been discussed in detail later in the manuscript (Sections 4.4 and 4.5). We have added a short explanation with respect to the above mentioned point within the manuscript.

Moreover, to elaborate on referee's comment, the dN/dlogD size distributions for Fig. 5 in the manuscript is shown in Fig. 1 here. In this illustration, considering the whole droplet size range, the distributions seem to be shifted between the instruments. However, it must be noted that it can be either due to the different sizing properties or due to the

inaccuracy of sample volume estimation which affects the DNCs. The main cause can not be confidently determined, but the different trends below and above ∼13 $\mu$m cutoff can be observed. Below ∼13 $\mu$m, the size distributions between the instruments do not match properly even allowing for a vertical shift due to different sample volume estimates or horizontal shift due to sizing bias. On the other hand, above the ∼13 $\mu$m cutoff, the size distributions seem to be similar especially between the two different PDI methods with a vertical shift which can be attributed to the different sample volumes. This observation can agree with the result of PDF plots comparison in Fig. 5 where it was found that the PDI can be less sensitive in detection of small droplets.

[Figure]

**Fig. 1** Comparison between cloud droplet size distributions collected with the VisiSize and the PDI probe (analysed with two different methods) during a measurement on 13th July 2019 15:36-15:51.

3. In Figure 10 especially in panel A the fit does not seem to be very good for large sizes. Why did you choose polynomial, not a different shape, and how many terms did you keep for the polynomial fit?

We agree with the referee's point regarding the fact that the polynomial fit performance was not satisfactory over the whole size range in panel (a). In fact, the fit curve was only added to emphasise a trend that seems to be present in the data, and there was no physical reason behind choosing it. In the revised manuscript, Fig. 10 is plotted using semi-logarithmic scale (see Fig. 2 here). Subsequently, a reciprocal fit ($\frac{a}{x+b}$) is also plotted to better show the trend in the DNC ratio as the mean droplet size increases.

[Figure]

**Fig. 2** The DNC ratio of the VisiSize to the PVC-PDI as a function of the arithmetic mean diameter size. Panels from left to right show the raw data (using the default sample volume for the VisiSize), the results after the SV correction and the $D_{min}$ cutoff, and the results after additionally applying the coincidence filter to the PDI data. Each circle represents a single measurement of ~15 min long. The reciprocal fit curves ($y = \frac{a}{x+b}$) and the 1:1 ratio are shown with the dashed and black horizontal lines, respectively.

4. For general presentation I would suggest increasing the size of some figures for better readability and decrease the amount of white space between them, such as in figure 13. It looks very cramped, but there is plenty of white space to be used between the panels.

   Following the referee's suggestion, all the figures in Sections 4.4 and 4.5 have been demonstrated in logarithmic scale to be able to better show the data points and trends. Moreover, the extra white spaces between the panels of each figure have been erased as much as possible.

5. – Equation 2: $\tau_{Tot}$ was not introduced.

   – Line 121-122: grammar (missing an "and" or something similar)

   – Line 316: typo: "because"

   We appreciate pointing out the above mistakes. They have been corrected in the revised manuscript.

**References**

Burnet, F. and Brenguier, J. L.: Comparison between standard and modified Forward Scattering Spectrometer Probes during the Small Cumulus Microphysics Study., 19, 1516–1531, 2002.

Chuang, P. Y., Saw, E. W., Small, J. D., Shaw, R. A., Sipperley, C. M., Payne, G. A., and Bachalo, W. D.: Airborne phase Doppler interferometry for cloud microphysical measurements, Aerosol Science and Technology, https://doi.org/10.1080/02786820802232956, 2008.

Cooper, W. A.: Effects of Coincidence on Measurements with a Forward Scattering Spectrometer Probe, Journal of Atmospheric and Oceanic Technology, 5, 823 – 832, https://doi.org/10.1175/1520-0426(1988)005<0823:EOCOMW>2.0.CO;2, 1988.

Lance, S.: Coincidence Errors in a Cloud Droplet Probe (CDP) and a Cloud and Aerosol Spectrometer (CAS), and the Improved Performance of a Modified CDP, Journal of Atmospheric and Oceanic Technology, 29, 1532 – 1541, https://doi.org/10.1175/JTECH-D-11-00208.1, 2012.

Lance, S., Brock, C. A., Rogers, D., and Gordon, J. A.: Water droplet calibration of the Cloud Droplet Probe (CDP) and in-flight performance in liquid, ice and mixed-phase clouds during ARCPAC, Atmospheric Measurement Techniques, 3, 1683–1706, 2010.

---

## Referee Report (RR1)

Review for "Cloud microphysical measurements at a mountain observatory: comparison between shadowgraph imaging and phase Doppler interferometry" by Mohammadi et. al.

The manuscript "Cloud microphysical measurements at a mountain observatory: comparison between shadowgraph imaging and phase Doppler interferometry" shows a study comparing two methods to size cloud droplets and explores the settings and methods of how these instruments should be used to yield matching results between the two methods. The authors used the VisiSize D30 as well as a Phase Doppler Interferometer during a mountaintop field campaign. These in-situ methods outside the laboratory can help to improve cloud measurements, as the reliability of various measurement techniques are uncertain and a thorough examination and evaluation of techniques and how to improve them is important for interpretation of collected data as well as the conducting of future measurements. Since my last review, the authors have addressed all my comments satisfactorily and the manuscript is sufficiently improved. Hence, I recommend publication.